# Mechanical Stretch Activates TRPV4 and Hemichannel Responses in the Nonpigmented Ciliary Epithelium

**DOI:** 10.3390/ijms24021673

**Published:** 2023-01-14

**Authors:** Mohammad Shahidullah, Nicholas A. Delamere

**Affiliations:** 1Department of Physiology, College of Medicine, University of Arizona, 1501 N Campbell Avenue, Tucson, AZ 85724, USA; 2Department of Ophthalmology and Vision Science, College of Medicine, University of Arizona, 1501 N Campbell Avenue, Tucson, AZ 85724, USA

**Keywords:** ciliary epithelium, hemichannel, TRPV4, TRPV1

## Abstract

Previously, we reported a mechanosensitive ion channel, TRPV4, along with functional connexin hemichannels on the basolateral surface of the ocular nonpigmented ciliary epithelium (NPE). In the lens, TRPV4-mediated hemichannel opening is part of a feedback loop that senses and respond to swelling. The present study was undertaken to test the hypothesis that TRPV4 and hemichannels in the NPE respond to a mechanical stimulus. Porcine NPE cells were cultured on flexible membranes to study effects of cyclic stretch and ATP release was determined by a luciferase assay. The uptake of propidium iodide (PI) was measured as an indicator of hemichannel opening. NPE cells subjected to cyclic stretch for 1–10 min (10%, 0.5 Hz) displayed a significant increase in ATP release into the bathing medium. In studies where PI was added to the bathing medium, the same stretch stimulus increased cell PI uptake. The ATP release and PI uptake responses to stretch both were prevented by a TRPV4 antagonist, HC067047 (10 µM), and a connexin mimetic peptide, Gap 27 (200µm). In the absence of a stretch stimulus, qualitatively similar ATP release and PI uptake responses were observed in cells exposed to the TRPV4 agonist GSK1016790A (10 nM), and Gap 27 prevented the responses. Cells subjected to an osmotic swelling stimulus (hypoosmotic medium: 200 mOsm) also displayed a significant increase in ATP release and PI uptake and the responses were abolished by TRPV4 inhibition. The findings point to TRPV4-dependent connexin hemichannel opening in response to mechanical stimulus. The TRPV4-hemichannel mechanism may act as a mechanosensor that facilitates the release of ATP and possibly other autocrine or paracrine signaling molecules that influence fluid (aqueous humor) secretion by the NPE.

## 1. Introduction

The ciliary epithelium is an unusual double-layer structure formed by two different epithelial cell types [1]. The epithelial bilayer covers the surface of the ciliary body. The cell type that faces the aqueous humor compartment is the nonpigmented ciliary epithelium (NPE). It has a highly invaginated basolateral surface, reflecting its specialization for solute and water transport, and it has a high Na,K-ATPase activity [2,3,4]. The NPE layer is positioned on top of the pigmented ciliary epithelium layer (PE) that faces the stroma, or interior, of the ciliary body. Gap junctions are abundant at the point where the apical surfaces of the PE and NPE make contact and there is convincing evidence that the two cell layers are coupled [5].

The ciliary epithelium is the site of aqueous humor production as well as a barrier that helps isolate the interior of the eye. Tight junctions between neighboring NPE cells restrict paracellular diffusion between the blood and aqueous humor compartment [6,7]. The NPE and PE are thought to function more or less as a unit to produce aqueous humor, ion transport mechanisms in the NPE and PE working in a coordinated manner to bring about the net flux of solute and consequent osmotic flow of water across the cell bilayer in a blood-to-aqueous direction [5]. Aqueous humor formation is an active process that requires energy in the form of ATP that fuels Na,K-ATPase activity. Na,K-ATPase provides the driving force by establishing ion gradients that are the driving force for the various secondary active transporters located at the basolateral surfaces of the NPE and PE [1]. It has been estimated that a ciliary epithelial cell transports the equivalent of 30% of its own volume every minute [8]. Accommodating the throughput of such a large amount of water passing through a cell would pose a challenge to the cell’s maintenance of its own volume. A mismatch between solute entry and exit would cause cell swelling or shrinkage.

Previously we reported functional connexin hemichannels on the NPE basolateral surface [9]. Later, TRPV4 was observed at the same location [10]. In the lens, these two mechanisms are thought to cooperate such that TRPV4 activation leads to hemichannel opening. TRPV4-dependent hemichannel opening is part of a feedback loop mechanism that enables the lens to sense and respond to swelling [10,11]. In the NPE hemichannel and TRPV4 cooperation would be highly significant in modulating aqueous humor secretion through release autocrine or paracrine signaling molecules, such as ATP, adenosine and melatonin. The present study was undertaken to determine whether TRPV4 and hemichannels might function as a mechanosensor in NPE cells.

## 2. Results

TRPV4 is abundant in the ciliary processes. By immunolocalization it appears almost exclusively in the aqueous humor-facing nonpigmented cell (NPE) layer of the epithelial bilayer (Figure 1). Because TRPV4 channel activation is mechanosensitive, we were interested in whether NPE cells respond to stretch. In order to apply a reproducible mechanical stimulus, NPE cells in primary culture were grown as monolayers on a flexible membrane that permitted the application of an axial stretch stimulus. Cells were subjected to 10% cyclic stretch at 0.5 Hz (1 cycle/2 s). The release of ATP into the medium by cells subjected to this stimulus for 1–10 min was significantly increased (Figure 2). The maximal response occurred at a stimulus duration of 2–5 min. To examine the role of TRPV4, cells were subjected to the 10% cyclic stretch stimulus for 2 min in the presence of a TRPV4 antagonist, HC067047 (10 µM). Under these conditions, there was no significant increase of ATP release (Figure 3A). However, 10 µM HC067047 itself produced a small increase in ATP release. At this concentration, HC067047 is thought to be reasonably selective for TRPV4 [11,12]. While we are not able to explain why HC067047 itself increased ATP release, it is not unusual for an antagonist to act as a partial agonist under certain circumstances. RN-1734 (10 µM), a different selective TRPV4 antagonist [13], prevented the ATP release response to the stretch stimulus but did not cause significant ATP release when added alone (Figure 3B). A different TRPV channel inhibitor A889425 (1 µM), which is selective for TRPV1 [14], had no detectable effect on the ATP release response to stretch (Figure 3C).

Aside from exocytosis, ATP release generally requires ATP-permeable channels. This is because of the large molecular size (MW 507) and negative charge of the ATP which precludes it diffusing across the lipid bilayer. Connexins are one of the five known families of ATP-permeable channels [15]. To test whether connexins play a role in the stretch responses, propidium iodide uptake was determined in cells subjected to stretch for various periods of time. Propidium iodide is known to enter cells via connexin hemichannels [9]. A significant accumulation of PI was observed in cells that were subjected to the 10% cyclic stretch stimulus (Figure 4). The time course of the PI uptake response to stretch was similar to the ATP release response (Figure 2). To focus further on the possible role of connexins, cells were exposed to a connexin mimetic peptide Gap 27 (200 µM, 2 h preincubation), which is known to inhibit connexin hemichannels [16]. In the presence of Gap 27 the increase of ATP release was prevented in cells subjected to the stretch stimulus (Figure 5A). The ability of Gap 27 to prevent ATP response to stretch suggests that ATP exit results from connexin hemichannel opening. Gap 27 also prevented a stretch-induced increase in propidium iodide uptake (Figure 5B).

Because the ability of HC067047 to prevent the ATP release response to stretch points to a role for TRPV4 activation, we tested whether a TRPV4 agonist would elicit ATP release and PI uptake responses in the absence of a stretch stimulus. This was the case. ATP release (Figure 6A) and PI uptake (Figure 6B) were both increased by the TRPV4 agonist GSK1016790A (10 nM) (Figure 6). Moreover, Gap 27 prevented the responses to the TRPV4 agonist.

The stretch responses are consistent with a mechanosensitive ATP release mechanism that involves TRPV4 as well as connexin hemichannel opening. To determine whether the mechanism could be activated by an osmotic swelling stimulus, cells were exposed to hypoosmotic medium (200 mOsm) in the absence or presence of HC067047 and simultaneously measured ATP luminescence (real-time ATP release measurement). Under these conditions ATP release more than doubled (Figure 7A) at 2 min. Importantly, the ATP release response to an osmotic stimulus was absent in cells exposed to the TRPV4 antagonist HC067047. The contribution of hemichannel opening to the ATP release was confirmed by studies on PI uptake. Cells subjected to a hypoosmotic stimulus displayed a doubling of PI uptake and the response was abolished by HC067047 (Figure 7B).

## 3. Discussion

Taken together, the results show a TRPV4-hemichannel mechanism that enables NPE cells to respond to a mechanical stimulus. TRPV4 was observed in the ciliary epithelium of the mouse eye, localized specifically to the NPE cell layer. TRPV4 is expressed in a wide variety of tissues, and while it was initially proposed as an osmosensor, it is generally mechanosensitive and responds to a range of stimuli including shear stress and cell swelling [12,17,18]. The evidence here suggests TRPV4 plays a role in connexin hemichannel opening and ATP release that occurs when the cells are subjected to a stretch stimulus. Stretch-induced ATP release was prevented by the TRPV4 channel antagonist HC067047, and by the connexin mimetic peptide Gap 27. The role of hemichannel opening in ATP release response to stretch also was evident as increase in the ability of propidium iodide to enter the cell. Mechanical stretch could have other long-term effects but because the duration of most studies was <10 min, it is highly unlikely that responses such as altered cell proliferation contributed significantly to the observed responses. In the absence of a stretch stimulus, qualitatively similar ATP release and PI uptake responses were observed in cells exposed to the TRPV4 agonist GSK1016790A. The findings are consistent with an earlier report of unpaired connexins capable of forming functional hemichannels at the basolateral surface of the NPE in the porcine eye [9]. This is not the first report of a link between TRPV4 activation and hemichannel opening. In the lens of the eye, we found previously that TRPV4-dependent hemichannel opening is part of a feedback loop mechanism that enables the lens to sense and respond to swelling, while a separate TRPV1-dependent feedback loop responds to osmotic shrinkage [11,19,20,21].

ATP release in response to a mechanical stimulus occurs in many cell types, including neurons, endothelial cells and epithelia [22], often for the purpose of purinergic signaling that activates a compensatory response [23]. Some feedback pathways may be ATP-dependent while other are activated by ADP, AMP or adenosine that result from the action of ecto ATPases. Interestingly, the concentration of ATP detected in the bathing solution peaked at 2 min then declined. This might reflect hydrolysis of extracellular ATP by ecto-ATPase enzymes. Ecto ATPase activity in the ciliary body is thought to be robust [24]. Due to ecto-TPase activity, a burst of ATP release is likely to give rise to a transient peak of ATP concentration followed by a decline. It should be noted that TRPV4-hemichannel mechanism is likely to be one of several different ATP release mechanisms in the ciliary body [25].

As might be expected for an ATP release process that responds to mechanical stretch, the NPE TRPV4-hemichannel mechanism was activated by osmotic swelling. Cells responded to a hypoosmotic stimulus by releasing ATP in a manner that was suppressed by TRPV4 inhibition. The ability of NPE cells to respond to a swelling stimulus may be functionally significant. The function of NPE is to secrete aqueous humor that supplies nutrients to the lens and cornea, which are specialized for transparency and are thus avascular. Additionally, aqueous humor secretion into the eye causes there to be an intraocular pressure of ~15 mm Hg, which helps maintain the roughly spherical shape of the globe and thus the appropriate corneal surface curvature necessary to focus light onto the retina. During the course of aqueous humor production, NPE cells experience a significant volume throughput as fluid enters the PE, crosses the cell bilayer and exits the basolateral side of the NPE. An amount of water roughly equivalent to the cell’s own volume passes through the cytoplasmic compartment every 3 min [8]. Almost certainly, the steady state flow is associated with minor fluctuations of fluid entry and exit, which would subject NPE cells to repeated episodes of mechanical stretch. We are not able to measure the mechanical forces experienced by the NPE cells in vivo. In the present study, the experimental protocol was to subject the cells to 10% cyclic stretch with a frequency of 1 cycle per 2 s.

It is possible that the NPE TRPV4-hemichannel mechanism has a role in the maintenance of cell size (volume), while the NPE and PE work in a coordinated manner to produce aqueous humor. The two cell layers are remarkably different in terms of volume regulation. NPE cells respond with a regulatory volume decrease when they are subjected to osmotic swelling, while PE cells do not [26]. PE cells have been observed to respond with a regulatory volume increase when they are subjected to osmotic shrinkage, while NPE cells do not. Going by solute transporter expression, the PE cell layer seems to be specialized for solute entry while the NPE layer appears specialized for solute exit [27,28]. Certainly, the rich basolateral expression of different Na,K-ATPase isoforms [2,29,30] and the elaborate infolding of its basolateral plasma membrane [4] gives the NPE the characteristic appearance of a secretory cell. If a mismatch between fluid entry to the PE and exit from the NPE were to cause cell swelling, the TRPV4-dependent release of ATP may trigger a compensatory response.

The TRPV4-hemichannel mechanism also might enable the NPE to respond to tissue distortion cause by a change of intraocular pressure. There is a hydrostatic pressure gradient across the ciliary epithelium [31]. Pressure on the aqueous side of the ciliary epithelium bilayer is several mm Hg higher than pressure on the stromal side. ATP release could feasibly be part of a feedback control mechanism that responds to a change in intraocular pressure due to imbalance between aqueous humor formation and drainage. Released ATP or its metabolic product adenosine would be available to act as an autocrine or paracrine agent to effect changes in ciliary epithelium function. Previous studies showed that extracellular ATP causes reduced aqueous humor secretion in isolated arterially perfused bovine eyes [32]. Other laboratories have shown that the ATP breakdown product adenosine increases the rate of aqueous humor outflow [33].

## 4. Methods

### 4.1. Reagents and Antibodies

Drugs and chemicals, including GSK1016790A, Gap 27, A88 SNAP, SNP, propidium iodide, trypsin EDTA, fetal bovine serum, newborn calf serum and DMSO, were purchased from Sigma (St. Louis, MO, USA). HC 067, 047 was purchased from Tocris Biosciences (Ellisville, MO, USA). All other chemicals, including those for making Krebs solution, were purchased from Sigma. Depending on their solubility, test agents were dissolved in water or in DMSO from freshly opened ampules. The knockout validated, rabbit polyclonal anti-TRPV4 antibody was purchased from Alomone Lab (Catalog # ACC-034; Jerusalem, Israel). The goat anti-Rabbit IgG (H+L) Alexa Fluor Plus 488 secondary antibody was purchased from Thermo Fisher Scientific (Catalog # 35552, Waltham, MA, USA). DAPI used for nuclear counterstaining was obtained from Thermo Fisher Scientific (Waltham, MA, USA). HEPES-buffered DMEM was purchased from Fisher Scientific (Gibco Mfg. No. 12430062; Pittsburgh, PA, USA).

### 4.2. Immunolocalization

Immunolocalization studies were carried out on intact ciliary body (CB) of porcine eye paraffin sections using an approach described earlier [34,35]. Porcine CBs were fixed in 10% neutral buffered formalin (NBF) for 24 h at 2–4 °C. Then the eyes were transferred to 70% ethanol at 2–4 °C for at least 24 h or until preparation of the paraffin blocks by a standard procedure. Sections, 7-microns thick, were cut and deparaffinized. Tissue sections were incubated at room temperature for 90 min in 10% goat serum in PBS (blocking buffer). Primary antibody directed against TRPV4 was added (1:100 dilution of the supplied stock solution) and the samples incubated for 24 h at 4 °C. Control specimens received no primary antibody but only the blocking buffer. The specimens were washed 3 times with PBS and incubated 90 min at room temperature with fluorescent secondary antibody (Alexa Fluor 488,1:200 dilution, in blocking buffer). Nuclear counterstaining was carried out by incubating with DAPI (1:100 in PBS, 300 nM) for 10 min at room temperature. Stained samples were covered with scanty amount of Dako Antifade Mounting Medium (Agilent Technologies, Inc., Santa Clara, CA, USA) and a cover slip was placed carefully over the sample, avoiding any bubble formation. Images were captured using a Leica DMI 6000 microscope (Leica Microsys-tems, Deerfield, IL, USA). Fluorescence excitation was achieved using 488 and 358 nm laser excitation wavelengths for Alexa Fluor 488 and DAPI, respectively.

### 4.3. Cell Culture

Fresh porcine eyes were obtained from the University of Arizona Meat Science laboratory. The use of animal tissue was approved by the University of Arizona Institutional Animal Care and Use Committee. NPE cells were isolated following a procedure described earlier [36] and grown in primary culture in HEPES-buffered DMEM containing 10% fetal bovine serum in a humidified incubator, saturated with 95% air and 5% CO_2_. Prior to use, the cell monolayers were kept in serum-free medium for 12 h. Then the medium was removed and replaced with Krebs solution at 37 °C containing (in mM) 119 NaCl, 4.7 KCl, 1.2 KH_2_PO_4_, 25 NaHCO_3_, 2.5 CaCl_2_, 1 MgCl_2_ and 5.5 glucose, equilibrated with 5% CO_2_ and adjusted to pH 7.4.

### 4.4. Stretch Stimulus

NPE cells were cultured to confluence on collagen IV-coated Flexible-bottomed BioFlex^®^ 6-well culture plates (Flexcel International Corporation, Burlington, NC, USA). The BioFlex^®^ 6-well plates were used with a BioFlex^®^ Baseplate Kit and a Flexcell^®^ Tension System (Flexcell^®^ FX-5000TM) to provide a mechanical stretch protocol to NPE monolayers maintained at 37 °C in a humid atmosphere at 5% CO_2_ by placing the apparatus in a tissue culture incubator. The cells were subjected to 10% cyclic stretch at 0.5 Hz (30 cycles/min) for 1, 2, 5 or 10 min. Stretch and control (no stretch) experiments were carried out simultaneously using cells derived from a single pool grown on identical plates.

### 4.5. Measurement of ATP

Based on the rationale that ATP and PI are able to pass through hemichannels, ATP release and PI uptake were measured as a two-part strategy, described earlier [9], to study hemichannel opening. After cells were subjected to a specified stretch stimulus, 100 µL of the Kreb’s solution was collected, snap frozen in liquid N_2_ and stored at −80 °C. ATP in the sample was measured with a bioluminescence assay based on the ATP-dependent luciferin–luciferase reaction (Molecular Probes, Inc., Eugene, OR, USA), quantified with a plate reader (Victor3 V, Perkin Elmer, San Jose, CA, USA). The assay was calibrated by means of a standard curve obtained from different concentrations of ATP dissolved in Krebs solution. The luciferin–luciferase reagent (Sigma, St. Louis, MO, USA) was also dissolved in Krebs solution. In experiments with hypoosmotic solution stimulus, a separate standard curve was created using ATP dissolved in hypoosmotic Krebs solution, since hypoosmotic solution influences luminescence reading. The ATP concentration in the bathing solution, as a measure of ATP release by NPE monolayers, was calculated as nanomoles per mg protein.

### 4.6. Propidium Iodide (PI) Uptake

Propidium iodide (PI, MW = 668.4) was added to the Krebs solution in the presence or absence of the test agent/agents at the beginning of stretch application and at a concentration of 25 µM. After cells were subjected to a specified duration of stretch stimulus, they were further incubated to complete a total of 30 min uptake time including the stretch duration. The cells were then washed 3 times with control Krebs solution and homogenized for 1 min (4 strokes of 15 s at 5 s intervals) using Misonix S3000 sonicator at a 6 W power setting (Misonix, New York, NY, USA) in 250 μL of distilled water. Protein in the sample was determined using a bicinchoninic acid assay [37] and PI was measured in 100 µL of homogenate transferred to a flat clear bottomed black plate using a SpectraMax iD5 Multi-Mode Microplate Reader (Molecular Devices, San Jose, CA, USA) by quantifying fluorescence at excitation and emission wavelengths of 535 nm and 617 nm, respectively. The control level of PI fluorescence was somewhat variable in experiments carried out using a different batch of cells on a different day. For this reason, we expressed PI results relative to the specific control for each set of experiment. The results are expressed as relative fluorescence/mg protein.

### 4.7. Statistical Analysis

The results are presented as the mean ± SEM of data obtained from a specified number of independent experiments. Statistical comparisons were made using one-way analysis of variance followed by Šídák’s multiple comparisons test. A probability (*p*) value of <0.05 was considered significant.

## 5. Conclusions

The findings point to a TRPV4-dependent connexin hemichannel opening in response to mechanical stimulus. We do not rule out the possibility that NPE reaction to stretch might involve other mechanisms in addition to TRPV4 and hemichannels but, under the present experimental conditions, TRPV4 and hemichannel blockers almost completely prevented stretch-induced ATP release and PI uptake responses. The TRPV4-hemichannel mechanism may act as a mechanosensor that facilitates the release of ATP and possibly other autocrine or paracrine signaling molecules that influence fluid (aqueous humor) secretion by the NPE.

## Figures and Tables

**Figure 1 ijms-24-01673-f001:**
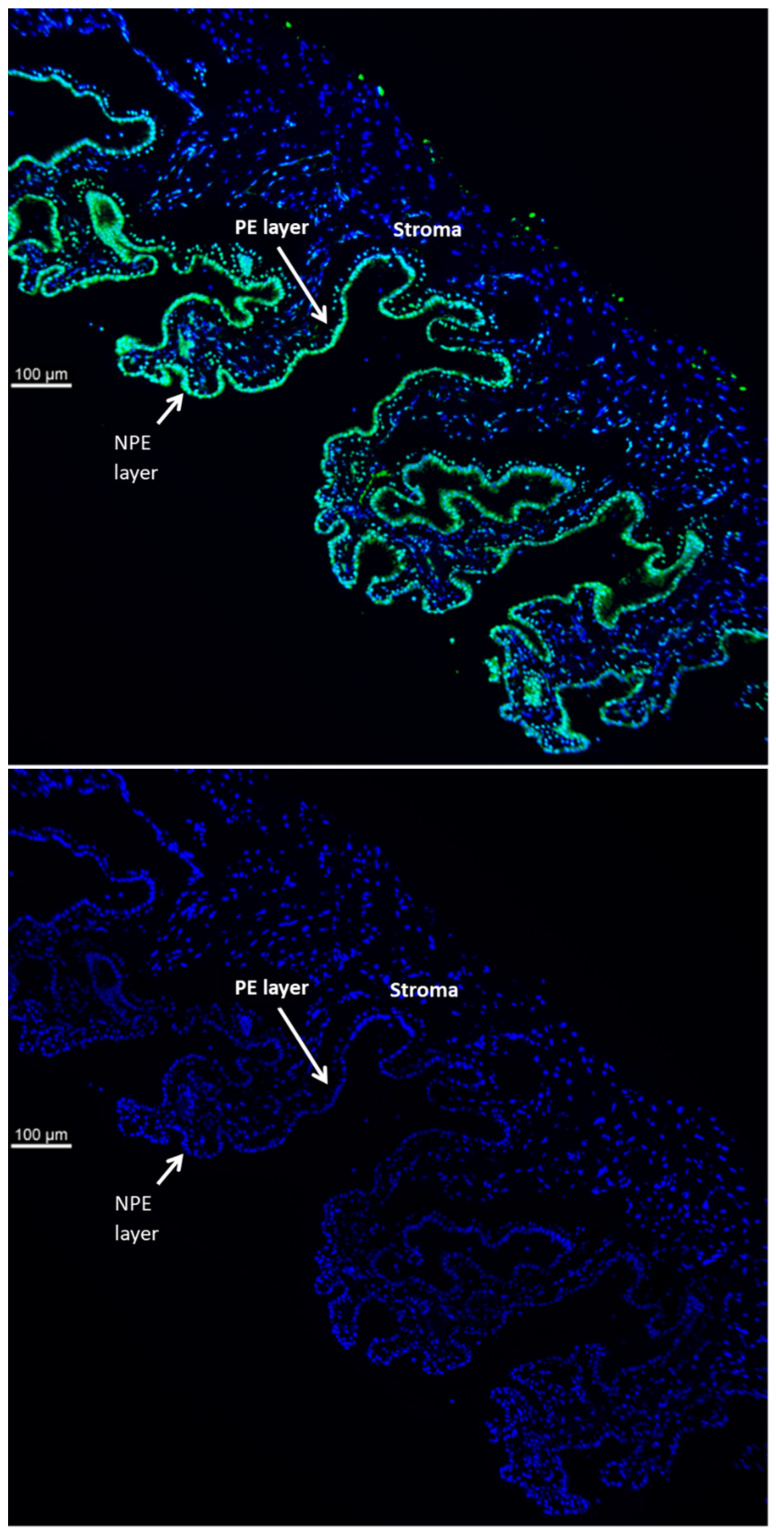
Immunolocalization of TRPV4 in the mouse ciliary body. TRPV4 (green) is abundant in the NPE (nonpigmented ciliary epithelium) cell layer at the surface but not the underlying pigmented ciliary epithelium (PE) cell layer or stroma. The lower panel shows a negative control in which no primary antibody was used.

**Figure 2 ijms-24-01673-f002:**
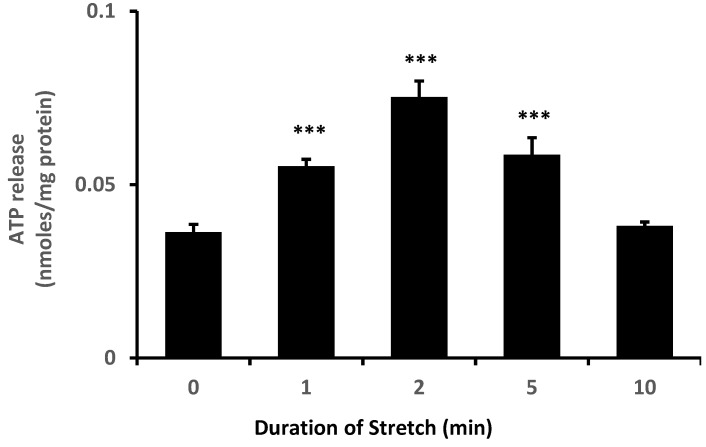
The influence of a stretch stimulus on ATP release. The amount of ATP in the bathing medium was measured after subjecting monolayers of cultured NPE cells to 10% cyclic stretch (0.5 Hz) for 1–10 min. The data are the mean ± SE of results from six independent experiments. *** (*p* < 0.001) indicates significant of differences from control.

**Figure 3 ijms-24-01673-f003:**
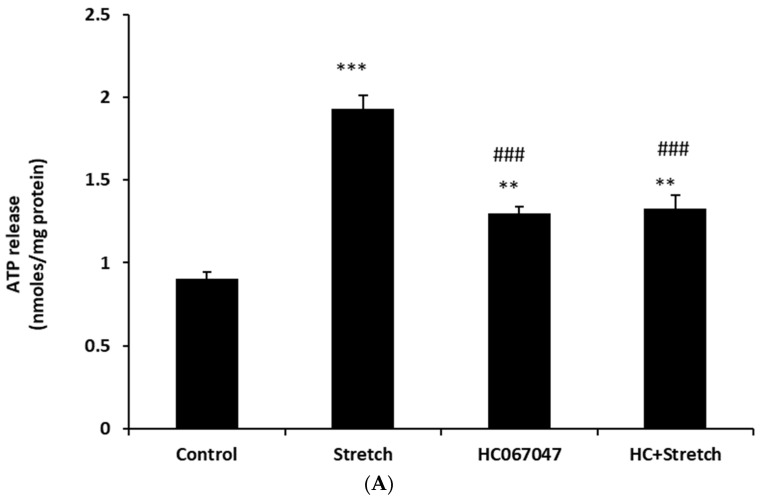
The ATP release response to cyclic stretch was inhibited by the selective TRPV4 antagonists HC 067047 and RN-1734 but not by the TRPV1 antagonist A889425. The amount of ATP in the bathing medium was measured after subjecting monolayers of cultured NPE cells to 10% cyclic stretch (0.5 Hz) for 2 min in the presence of either HC 067047 (10 µM) (**A**) or RN-1734 (10 µM) (**B**) or A889425 (1 µM) (**C**) added 20 min beforehand. The data are the mean ± SE of results from six independent experiments. ** (*p* < 0.01) and *** (*p* < 0.001) indicate significant differences from control and ### (*p* < 0.001) indicates significant differences from a stretch stimulus.

**Figure 4 ijms-24-01673-f004:**
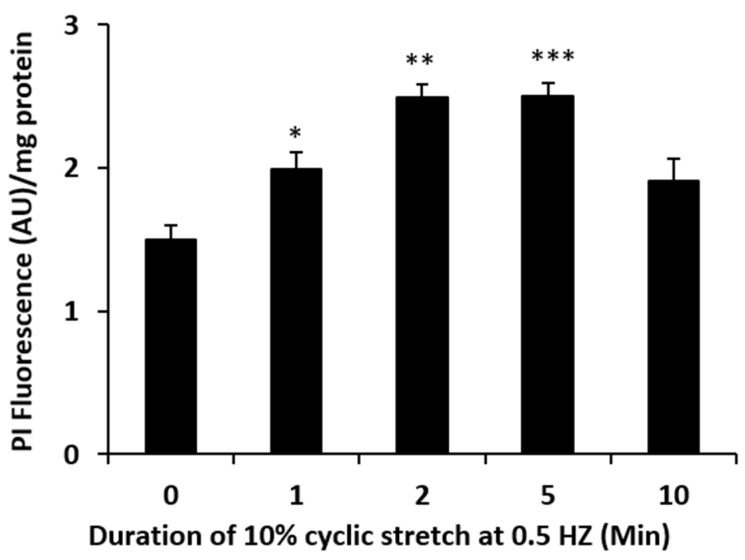
The influence of a stretch stimulus on the uptake of propidium iodide (PI), used here as an indicator of hemichannel opening. The results show PI fluorescence determined in monolayers of cultured NPE cells measured after subjecting the cells to 10% cyclic stretch (0.5 Hz) for 1–10 min in Krebs’ solution containing 25 µM PI. The fluorescence data, presented as arbitrary units (AU), are the mean ± SE of results from six independent experiments. * (*p* < 0.05), ** (*p* < 0.01) and *** (*p* < 0.001) indicate significant differences from control.

**Figure 5 ijms-24-01673-f005:**
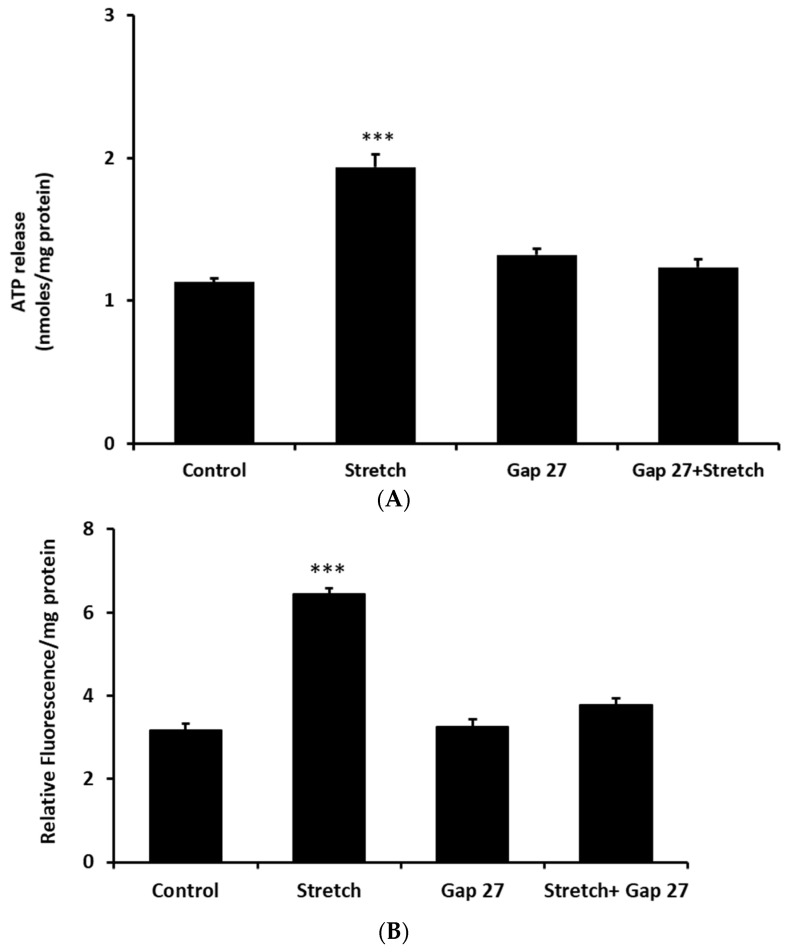
The ATP release and propidium iodide uptake responses to cyclic stretch were abolished by the connexin mimetic peptide Gap 27. The amount of ATP in the bathing medium and PI in cell lysate were measured after subjecting the monolayers of cultured NPE cells to 10% cyclic stretch (0.5 Hz) for 2 min in the presence of Gap 27 (200 µM) added 60 min beforehand. (**A**) shows effect on ATP response and (**B**) shows effect on PI uptake response. The data are the mean ± SE of results from six independent experiments. *** (*p* < 0.001) indicates significant differences from control.

**Figure 6 ijms-24-01673-f006:**
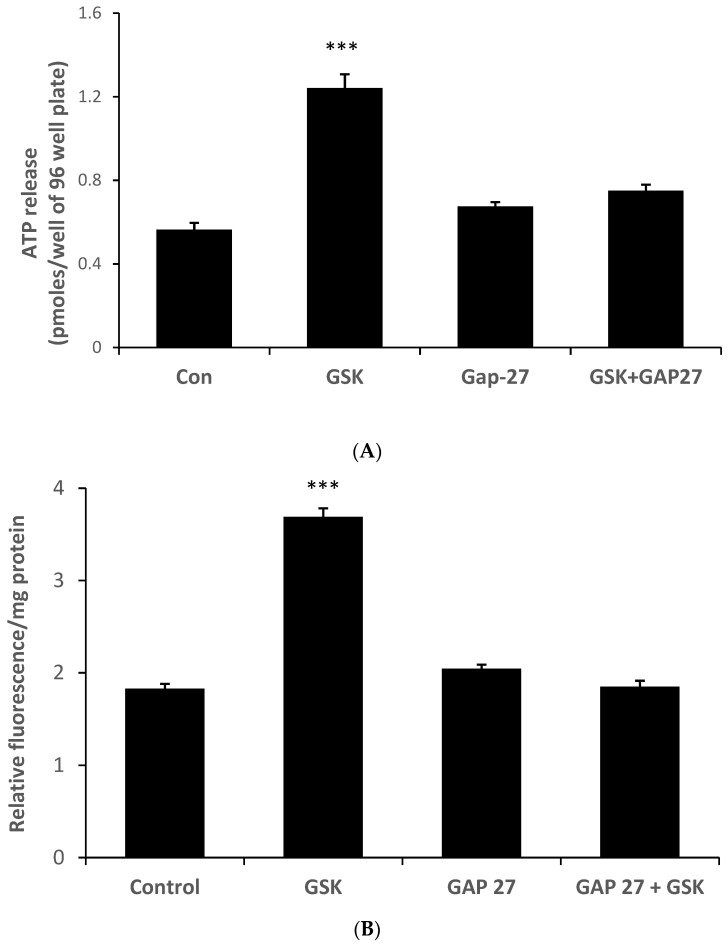
The connexin mimetic peptide Gap 27 prevents ATP release and PI uptake responses to TRPV4 activation by a selective agonist, GSK1016790A. The amount of ATP in the bathing medium (**A**) or cell PI fluorescence (**B**) was measured after exposing the monolayers of cultured NPE cells to GSK1016790A (10 nM) in the presence or absence of Gap 27 (200 µM) added 60 min beforehand. The data are the mean ± SE of results from six independent experiments. *** (*p* < 0.001) indicates significant differences from control.

**Figure 7 ijms-24-01673-f007:**
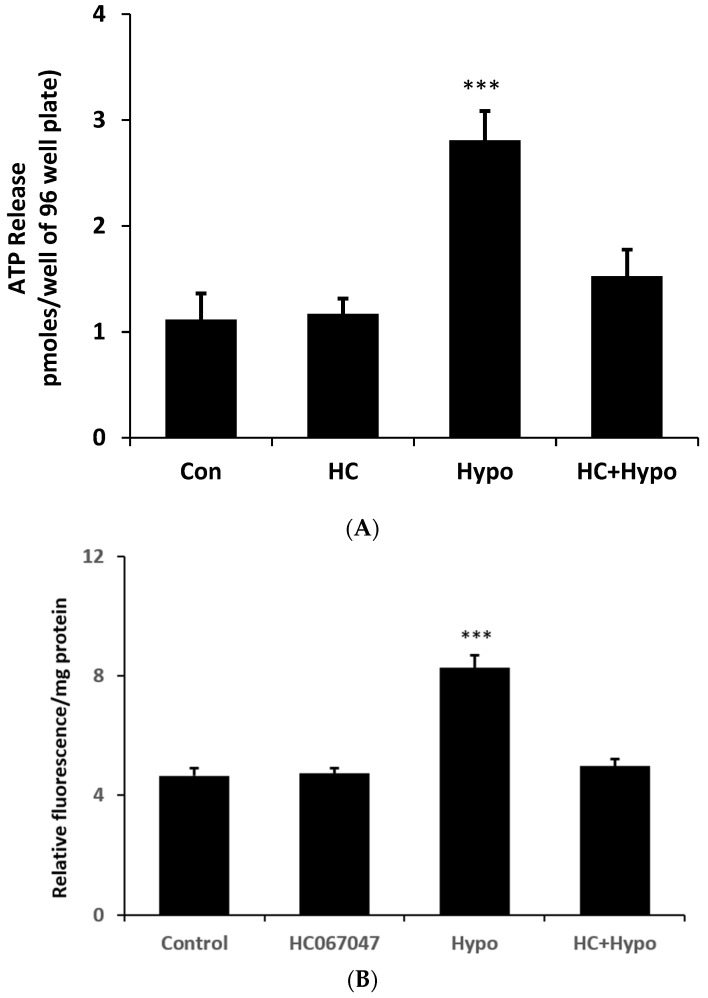
The TRPV4 antagonist HC067047 prevents ATP release and PI uptake responses to an osmotic swelling stimulus. The amount of ATP in the bathing medium (**A**) or cell PI fluorescence (**B**) was measured after exposing the monolayers of cultured NPE cells to hypoosmotic Kreb’s solution (200 mOsm) for 2 min in the presence or absence of HC067047 (10 µM) added 20 min beforehand. The data are the mean ± SE of results from twelve (**A**) or six (**B**) independent experiments. *** (*p* < 0.001) indicates significant differences from control.

## Data Availability

Generated and analyzed data for this study can be found in our data repository at: https://arizona.box.com/s/zfp2m030kd9kmpd4iuqaxd27b1dqcehx (accessed on 12 January 2023).

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
