# Peer review of "Mechanical Stretch Activates TRPV4 and Hemichannel Responses in the Nonpigmented Ciliary Epithelium"

_ijms, 2023, doi:10.3390/ijms24021673_

Round 1

Reviewer 1 Report

Please check the attached file

Author Response

A Pdf file attached.

Reviewer 2 Report

The article reveals an interesting information about mechanosensitivity  of TRPV and hemichannels in NPE. I have minor suggestions:

1) Running title of an article does not represent the main idea and does not provide information about mechanosensitivity

2) Color scheme on the graphs makes them a little bit unreadable (especially error bars)

Author Response

A Pdf file is attached.

Round 2

Reviewer 1 Report

Thanks for the authors addressing my comments, I accepted their edition and clarification, I do not have further comments for publication.